# Natural Product-Derived Phosphonic Acids as Corrosion Inhibitors for Iron and Steel

**DOI:** 10.3390/molecules27061778

**Published:** 2022-03-08

**Authors:** Erik Ruf, Tim Naundorf, Tom Seddig, Helmut Kipphardt, Wolfgang Maison

**Affiliations:** 1Department of Chemistry, Universität Hamburg, Bundesstraße 45, 20146 Hamburg, Germany; ruf@chemie.uni-hamburg.de (E.R.); naundorf@chemie.uni-hamburg.de (T.N.); seddig@chemie.uni-hamburg.de (T.S.); 2Metall-Chemie Technologies GmbH, Kaiser-Wilhelm-Straße 93, 20355 Hamburg, Germany; kipphardt@metall-chemie-technologies.com

**Keywords:** corrosion, corrosion inhibitors, phosphonic acids

## Abstract

Organic acids, typically derived from an oil-based value chain, are frequently used as corrosion inhibitors in industrial metal working fluids. The criteria for selection of these corrosion inhibitors have changed in the last decades, and are today not only performance-driven, but influenced by ecological considerations, toxicity and regulatory standards. We present scalable semisynthetic approaches to organic corrosion inhibitors based on phosphonic acids from renewable resources. They have been evaluated by chip filter assay, potentiodynamic polarization measurements, electrochemical impedance measurements and gravimetry for corrosion protection of iron and steel in an aqueous environment at slightly alkaline pH. The efficacy of several phosphonic acids tested was found to be strongly dependent on structural features influencing molecular self-assembly of protective layers, and the solubility of salts formed with di- and trivalent cations from the media or formed during corrosion. A carboxyphosphonic acid (derived from castor oil) was found to have remarkable anticorrosive effects in all media tested. We attribute the anticorrosion properties of this carboxyphosphonic acid to the formation of particularly stable protective layers on the metal surface. It might thus serve as a commercially attractive substitute for current acidic corrosion inhibitors, derived from renewable resources.

## 1. Introduction

Corrosion-induced damage not only poses high security risks, but is also responsible for financial losses of around 2.5 billion USD worldwide, according to NACE (National Association of Corrosion Engineers) [1]. Among many other areas, manufacturing processes involving iron and steel are often compromised by corrosion [2]. Metal working fluids (MWFs) have therefore had a major role in manufacturing processes since pre-industrialized times. They are usually oil- or water-based emulsions or solutions, with a high number of additives providing desired properties such as anti-wear, anti-foam, and antimicrobial features as well as corrosion inhibition [3]. Depending on the field of application, various compounds with anti-corrosive properties for iron and steel have been used. The criteria for selection of these corrosion inhibitors have changed significantly in the last decades, and are today not only performance-driven, but influenced by ecological considerations, toxicity and regulatory standards [3]. Water-based MWFs of neutral or slightly alkaline pH containing ingredients of low volatility are particularly attractive in this context. A widely used class of substances for aqueous MWFs are acidic corrosion inhibitors such as carboxylic [4] or phosphonic acids [5]. In aqueous media, these compounds may be used either for applications at low pH [6] or at neutral and alkaline pH [7]. The mechanisms of action of these acidic corrosion inhibitors are heterogenous, often not well understood, and are highly dependent on the molecular structure of acid derivatives [8,9]. Metal surface binding is typically an important factor for corrosion protection with amphiphilic acids [4,10,11,12]. Phosphonic acids have been frequently used as layer-forming anticorrosives, because phosphonates show a particularly strong chemisorption on numerous metals relevant for industrial applications [13,14]. Their ability to form stable self-assembled monolayers (SAMs) has been well-documented in the literature, and they have been used for many applications besides corrosion inhibition [15,16]. In contrast to other substance classes, this ability is not limited to stainless steel or iron [17], but extends to many other metals, such as aluminum, magnesium, titanium, zirconium and the corresponding oxides [18]. Various phosphonic acids have therefore been used in large quantities as additives in MWFs. Among them are amphiphilic alkylphosphonic acids [19,20]. In addition, more polar amino- or hydroxy-phosphonic acids are often found in MWFs with antiscalant properties [21]. All of these compounds are typical industrial products from an oil-based value chain. As mentioned above, there is an increasing need for environmentally friendly corrosion inhibitors, which ideally can be obtained from renewable resources. In this context, intensive research has been carried out to find new corrosion inhibitors among plant-based bio-materials [22], natural product mixtures [23] or pure natural compounds [24] as well as specifically synthesized organic molecules intended to be benign by design [25,26,27]. In some aspects, this search for active substances in natural resources has parallels to the drug development process in pharmaceutical industry, where new lead compounds are frequently natural products. Often, these lead compounds are slightly modified via organic synthesis to become successful drugs on the market. In this work, we present the application of this general semisynthetic approach to the development of new corrosion inhibitors for iron and steel via synthesis and evaluation of natural product-derived phosphonic acids [28]. We focus on derivatives which are available in one or two scalable synthetic steps from readily available and renewable feed stocks, such as plant-derived oils and extracts.

## 2. Results

### 2.1. Synthesis

Our target compounds were designed to be nonvolatile alkylphosphonic acids with high chemical stability, amphiphilic character and good surface activity. In addition, they should be readily available in one or two scalable synthetic steps from cheap precursors derived from renewable resources, such as plant extracts or oils. A number of methods for the synthesis of alkylphosphonic acids have been reported in the literature, and the reader is referred to the recent review literature for an overview [16,18,29]. However, many methods rely on multistep procedures and/or have other drawbacks, such as low atom economy, use of expensive reagents or hazardous protocols. They are thus of limited use for upscaling and industrial applications. Herein, we focus on scalable one- or two-step modifications of alkenes or alcohols to phosphonic acids. These functional groups are present in various plant-derived natural products, such as mono- and sesquiterpenes, as well as fatty acids. A number of these natural products are available in large tonnages, either directly from natural sources or via simple industrial processes, such as pyrolysis of plant extracts or oils [30].

In a first attempt to synthesize amphiphilic phosphonic acids from natural products, we focused on a one-pot protocol established by Bravo-Altamirano and Montchamp [31]. The method is suitable for the conversion of terminal alkenes and non-hindered allylic alcohols to the corresponding phosphonic acids [32,33]. It consists of a Pd-catalyzed hydrophosphinylation and subsequent oxidation of an intermediate phosphinic acid derivative in the same reaction vessel. It is thus one of the rare direct phosphonylation reactions using hypophosphorous acid. From a practical point of view, reaction control is easy via ^31^P-NMR, because reactant (hypophosphorous acid), intermediate (alkylphosphinic acid), product (alkylphosphonic acid) and by-product (H_3_PO_4_) can be detected. We have applied this protocol to the synthesis of phosphonic acid (**3**) from undecylenic acid (**1**)**,** as outlined in Figure 1. The reaction proceeded smoothly via intermediate phosphinic acid (**2**), which was not isolated, but immediately oxidized with hydrogen peroxide to the final product (**3**) in almost quantitative yield. We scaled this reaction up to ~0.5 mol without any problems. The oxidation of intermediate phosphinic acids, such as **2,** can typically be performed in the same vessel just by heating the reaction mixture under air. However, in large-scale conversions, the addition of a stoichiometric oxidant, such as hydrogen peroxide, gave much faster conversions. We did not attempt to optimize the upscaling of this procedure in this study. However, recent studies indicate that the catalyst loading in the first step could be significantly lowered, and that the second oxidation step could be performed under air if sufficient mixing of the solution can be guaranteed [34]. The direct Pd-catalyzed one-pot conversion of undecylenic acid to carboxyphosphonic acid (**3**) is thus more efficient than comparable two-step procedures [30,35,36].

The protocol is also applicable to selected allylic alcohols, as exemplified by the conversion of geraniol (**4**) in Figure 1. In this case, the intermediate phosphinic acid (**5**) was oxidized with DMSO and catalytic iodine to produce the final phosphonic acid (**6**). We have also applied the protocol to other natural products, available at low cost. A few of the resulting phosphonic acids (**7**–**9**), derived from camphene, isopulegol and β-pinene, are depicted in Figure 1. Yields of these products dropped significantly, compared to **3** or **6**, either because of steric hindrance or competing side reactions. Substrates with internal double bonds were particularly problematic: oleic acid, for example, with an isolated internal double bond, gave only low amounts of phosphonylation products (two regioisomers, according to ^1^H-NMR analysis of the crude product), even after long reaction times (6 days). We have not tried to purify the resulting phosphonic acids obtained by this protocol (vide infra for an alternative procedure), and therefore did not include the data here. Other terpene starting materials with more than one alkene functionality, such as myrcene, valencene and limonene, gave only complex product mixtures. Obviously, various competing side reactions occur with these diene precursors besides (unselective) phosphonylation of double bonds.

The abovementioned one-pot protocol, consisting of a Pd-catalyzed hydrophosphinylation and subsequent oxidation, is an efficient and mild method for the synthesis of phosphonic acids from fatty acids and terpenoids. However, it is restricted to a few commercially attractive starting materials for these natural product classes, such as undecylenic acid and geraniol. For less reactive, but economically attractive substrates such as β-pinene, we obtained rather low yields of the corresponding phosphonic acid (**9**), and have therefore explored the direct radical hydrophosphinylation of β-pinene as a workaround [37]. We used the conditions outlined by Deprele and Montchamp [38] to convert β-pinene (**10**) with ammonium hypophosphite (Figure 2). In our experiment, the subsequent oxidation of the intermediate phosphinic acid (**11**) with air was quite slow. Again, we used a DMSO/iodine-mediated protocol to achieve a faster oxidation to the corresponding phosphonic acid (**12**). The reaction produced the unexpected phosphonic acid (**12**) as the main product, which was clearly not identical with phosphonic acid (**9**) obtained by Pd catalysis above (Figure 1). A possible mechanistic rationale for the observed major product **12** is shown in Figure 2, and is in accordance with earlier observations for other radical reactions of β-pinene [39,40].

We were also interested in the synthesis of alkyldiphosphonic acids from natural product precursors with two double bonds. For this purpose, we used more forcing reaction conditions and a two-step approach of radical phosphite addition to alkenes, followed by hydrolysis of the resulting phosphonate esters. This method has been reported for the conversion of limonene by Biresaw et al. In this case, phosphonates were obtained by treatment of limonene with dialkylphosphites and a radical initiator [41]. The process gave a complex mixture of mono- and diphosphonic acid esters, each as isomeric mixtures. However, for technical applications, this lack of selectivity might be tolerable. We applied slightly modified reaction conditions to limonene and valencene as starting materials. Both are cheap by-products from industrial processing of citrus fruits such as lemons and oranges. We also converted β-pinene due to our earlier observation that an additional double bond was formed during the first phosphorylation reaction (Figure 3). In all cases, diphosphonic acids (**14**–**16**) were obtained in good yields, but as complex isomeric mixtures. We did not attempt to separate these isomers, and have therefore not fully characterized these products. The ^1^H and ^13^C-NMR spectra of these isomeric mixtures are quite complex. In addition, the mixtures were obtained as hygroscopic oils or sticky solids, making elemental analysis difficult. However, HRMS clearly confirmed the elemental formula and ^31^P-NMR indicated only phosphonate signals around 30 ppm (see Appendix A for analytical details). Isolation and purification of intermediate phosphonate esters is advantageous before final hydrolysis to phosphonic acids (**14**–**16**). The latter are hard to purify by chromatography or crystallization.

The same forcing conditions were used for the conversion of oleic acid and ricinolic acid, both cheap starting materials available from natural oils (Figure 4). For oleic acid (**17**), the two-step procedure gave a clean conversion to the corresponding phosphonic acid (**18**), which was obtained as a racemic mixture of two isomers (only one regioisomer shown). The reaction is scalable, and has been performed up to 50 g scale. Ricinolic acid (**19**) gave an unexpected reaction product (**24**). The ^31^P-NMR spectrum of **24** shows eight distinct peaks, clustered in two groups of four at 27–51 ppm. We attributed the formation of these eight isomers of **24** to an initial transesterification of **24** to **20**, which was likely followed by elimination to **21**, leaving four different sites for phosphorylation to intermediates (**22**) (only one of four possible isomers shown). Final isomerization to product **23** adds two double bond isomers to each regioisomer, resulting in a total of eight isomers. We did not attempt to separate these isomers in the final product (**24**) after hydrolysis, but rather used the mixture in corrosion tests.

To complement our set of amphiphilic monoterpene-derived phosphonic acids, we converted citronellol (**25**) to phosphonic acids **27** and **28** (Figure 5). In a first step, citronellol (**25**) was converted to the corresponding bromide by Appel reaction, followed by a Michaelis–Arbuzov reaction to ester **26**. This intermediate was converted to unsaturated phosphonic acid (**27**) by deprotection with bromotrimethylsilane. Alternatively, ester **26** was hydrogenated and deprotected to the saturated phosphonic acid (**28**). Although this synthetic approach is certainly not economically viable, the set of three geraniol- and citronellol-derived phosphonic acids (**6**, **27** and **28**) with 0–2 double bonds, might be useful in deciphering the role of unsaturation for layer formation on metals. It is also notable that phosphonic acid derivatives of geraniol and citronellol have received some interest as inhibitors of prenyltransferases for pharmaceutical applications [42,43]. 

### 2.2. Evaluation of Corrosion Inhibition on Steel

With a range of phosphonic acids in hand, we evaluated their efficiency for corrosion protection of iron and steel. In this study, we focused on corrosion protection in aqueous solution at slightly alkaline pH values. These are common conditions for the application of MWFs in various metal-processing industrial processes. For this purpose, our phosphonic acids were neutralized with triethanolamine (TEA, a standard alkaline additive for acidic corrosion inhibitors) in water or saline, and the protective effect was compared to commercial octylphosphonic acid (**OPA**) and 1,3,5-triazine-2,4,6-triaminocaproic acid (**TC**). The latter is a tricarboxylic acid used in many aqueous industrial MWFs, and is thus used as a benchmark corrosion inhibitor for iron and steel [44]. We are aware of the fact that MWFs are usually complex mixtures containing various additives (vide supra), which often have an additional impact on the overall performance of the MWFs in metal processing. However, a complete evaluation of MWFs is beyond the scope of this study, which rather focusses on the anticorrosive effects of the isolated phosphonic acids. We have used three different types of assays to evaluate the phosphonic acids: 1. a grey cast iron chip filter assay according to DIN 51360-2, 2. electrochemical analysis on steel, and 3. a long-term gravimetric corrosion assay on steel.

#### 2.2.1. Chip Filter Assays

The chip filter assay according to DIN 51360-2 is a standard method used in industry for fast evaluation of the anticorrosive properties of MWFs for iron. Briefly, a defined amount of sieved grey cast iron chips is placed on a round paper filter and submerged in a hard water solution of the neutralized (here with TEA as a base) acidic corrosion inhibitor at the appropriate concentration (Appendix A, see Appendix A). After incubation and subsequent washing of the filter, corrosion marks are visually detected and scored between 0 (no corrosion marks) and 4 (strong staining of the filter due to corrosion). This assay is operationally very simple and can be easily used for medium throughput screening. In addition, it is quite sensitive, due the large surface area and the high corrosivity of grey cast iron chips. However, the optical detection of staining is slightly subjective, and misinterpretation according to staining that is not caused by corrosion is possible. We have therefore used this assay for a fast preselection of corrosion inhibitors. Successful derivatives were afterwards submitted to more detailed electrochemical evaluation and a long-term gravimetric assay. We tested all synthesized phosphonic acids at different concentrations to obtain limiting effective concentrations (Appendix A, see Appendix A). Inhibitors similarly effective to **TC** (scores 0 and 1 at 10 mmol/L or lower) were considered for further evaluation. Phosphonic acids (**3**, **7**, **12** and **18**) were found to give low scores at concentrations around 10 mmol/L. Assuming a layer-forming anticorrosive mechanism, the observed cutoff for inhibitor efficiency at 5–10 mmol/L matches the observed critical micelle concentrations (CMC), at least for **TC** (CMC_(**TC**-TEA)_ = 8.9 mmol/L), **OPA** (CMC_(**OPA**-TEA)_ = 4.7 mmol/L), **3** (CMC_(**3**-TEA)_ = 4.7 mmol/L) and **7** (CMC_(**7**-TEA)_ = 4.7 mmol/L), which we determined for the TEA salts by ^1^H-NMR in water [45]. It is notable that **TC** and the carboxyphosphonic acids **3** and **18** were compatible with the presence of secondary metal cations in hard water. **OPA**, like all other phosphonic acids tested, was hardly compatible with secondary metal cations, such as Ca^2+^ and Mg^2+^ from the hard water test media, and produced milky suspensions when applied at 10 mmol/L. The performance of several amphiphilic phosphonic acids might thus be compromised by the limited solubility of salts formed with divalent cations. Diphosphonic acids (**14**–**16**) were completely incompatible with hard water (20 dGH) above concentrations of 1 mmol/L, leading to gel formation. Phosphonic acids **6**, **27** and **28** were found to be slightly less effective than **3** and **18**, providing scores of 0 or 1 at concentrations higher than 20 mmol/L only.

#### 2.2.2. Electrochemical Analysis

The time-dependent variation of OCP of S235JR steel in hard water (20 dGH) for the selected carboxyphosphonic acids **3** and **18** as well as **TC** and **OPA** for comparison is shown in Figure 1. Amphiphilic phosphonic acids have been shown to form corrosion-protective layers on metal surfaces, as mentioned above. This process is driven by chemisorption of the phosphonic acid to the metal surface, and packing effects of the side chain such as van der Waals interactions or hydrogen bonding, and is thus dependent on the structure of the phosphonic acid. In most reported cases, the assembly of amphiphilic phosphonic acid layers on iron or steel is a slow process over several hours to days [46]. This correlates well with the time-dependent efficacy of many phosphonic acids as corrosion inhibitors on iron and steel, as measured by electrochemical methods [46]. We followed OCPs over a time span of 24 h. For all solutions containing inhibitors, OCPs were significantly lower in value at any time compared to the reference without inhibitor, reflecting the formation of protective layers on the metal surface [47]. Both carboxyphosphonic acids **3** and **18** show an initial decrease in values over a few hours, followed by constant low OCPs. This is in accordance with the formation of protective layers on the metal surface, which are the reason for the known anodic inhibition of phosphonic acids, and parallels observations by Kalman et al. for long-chained ω-diphosphonic acids [48]. It is notable that **OPA** does not show a comparable decrease, but rather a slight increase in OCP values before reaching a constant value after 10 h. It has been described that **OPA** layers have a limited stability under alkaline conditions, in contrast to phosphonic acid layers of longer chain length [14]. However, the inhomogenous increase seen in the first hours might reflect poor compatibility of **OPA** with hard water, leading to precipitation of **OPA** salts with divalent cations, and thus concentration levels below surface aggregation concentration (as a rough estimate, the critical micelle concentration of **OPA**/TEA salt in water was measured by NMR: CMC_(**OPA**-TEA)_ = 4.7 mmol/L, see Appendix A). A less-perfect layer formation compared to the carboxyphosphonic acids **3** and **18** is the consequence, along with low levels of corrosion protection. The observed effect is clearly dependent on the choice of media: If **OPA** is applied in demineralized water (pH adjusted to 8.5 with 3 eq TEA), no precipitation is observed, and the OCP (see Figure 1A, brown line) shows the expected decrease in value with time, in accordance with the literature [49]. Interestingly, OCP values of **18** also do not constantly remain low, but show a significant increase to a value similar to that of **OPA** after 16 h, which also reflects the destruction of a quality protective layer at this time. In parallel, a precipitate was formed, leading to development of a milky suspension. This precipitation might again be the reason for a lack of regeneration of the protective layer, due to low concentration of **18** in solution. We assume that the layers formed by carboxyphosphonic acid **18** are less stable than layers formed by **3**. This might reflect less-ordered packing of side chains upon self-assembly, caused by the branched structure of **18**. Accordingly, in comparison to **3** (CMC_(**3**-TEA)_ = 4.7 mmol/L) and **OPA** (CMC_(**OPA**-TEA)_ = 4.7 mmol/L), we have observed a significantly higher CMC for carboxyphosphonic acid **18** (CMC_(**18**-TEA)_ = 19.1 mmol, see Appendix A). The tricarboxylic acid **TC** rapidly establishes a constant OCP, reflecting the fast formation of protective layers on steel. A mechanistic investigation of **TC** and close derivatives has, to the best of our knowledge, been performed in acidic media only, making extrapolation to alkaline systems difficult [44]. However, as a polar tricarboxylic acid derivative of heteroaromatic triazine, it most likely has complex adsorption properties on metal surfaces, with contributions from the carboxylic acids, amines and the heteroaromatic system.

Potentiodynamic polarization curves were obtained at room temperature for S235JR steel after 24 h exposure to different test media containing the corrosion inhibitors and TEA as alkaline additive (pH of test solutions 7.9–8.8). In a first set of experiments, we evaluated phosphonic acids **3**, **18** and **OPA** along with **TC** in hard water (20 dGH), and a selection of the resulting parameters is presented in Table 1. To verify our concerns with respect to the hard water stability of **OPA**, we also measured the potentiodynamic polarization curves and EIS of **OPA** in demineralized water of pH 8.5 (data included in Figure 1 and entry 9 in Table 1). We measured the two most potent inhibitors, **TC** and **3**, in different concentrations, and observed a limiting effective concentration for **TC** of 2.2 mmol/L and for carboxyphosphonic acid **3** of 3.8 mmol/L. Above these concentrations, almost complete inhibition of corrosion was observed (*IE_corr_* > 99%). For comparison, data for 64.4 mmol/L (**TC**) and 112 mmol/L (**3**) are also shown. Both correspond to 3 wt% of the appropriate CI, which is a common concentration of commercial CIs such as **TC** in industrial MWFs. A concentration of 18.8 mmol/L (corresponding to 0.5 wt% of **3**) was used for comparison of the three phosphonic acids **3**, **18** and **OPA**. At this concentration, corrosion marks are visible on the test specimen after 24 h for treatments with **18** and **OPA**. A selection of the corresponding polarization curves and the Bode plots from EIS measurements are depicted in Figure 1. Together with the data in Table 1, these analyses confirm our earlier findings from the chip filter assay with grey cast iron: **TC** and all of our tested phosphonic acids show good to excellent corrosion protection on steel. All acidic corrosion inhibitors led to an increase in corrosion potential (*E_corr_*) and a decrease in corrosion current density (*j_corr_*), along with significantly decreased corrosion rates. However, after 24 h, a difference in efficiency was observed among the tested inhibitors: **TC** and carboxyphosphonic acid **3** show superior inhibition efficiency, even at lower concentrations (entries 3 and 5 in Table 1). Carboxyphosphonic acid **18** and **OPA** are slightly less effective (entries 7 and 8 in Table 1). This might be explained with the relatively low hard water solubility of **OPA**, and the less-perfect layer assembly of **18**, as mentioned above. The electrochemical data for **OPA** support this hypothesis. Consequently, in pH 8.5, water-stable layer formation and excellent anodic corrosion protection were observed for **OPA**. Bode plots of electrochemical impedance measurements confirm the high efficiency of **TC** and carboxyphosphonic acid **3** as layer-forming corrosion inhibitors, because the values of the Bode modules in the low-frequency region are highest for these two compounds (Figure 1C). Their capacitive loops in the Nyquist plots are also the largest among all compounds tested (see Appendix A, Appendix A). Both high Bode moduli and large capacitive loops in Nyquist plots correlate with high corrosion resistance [50,51]. The imperfect layer formation of **18** is reflected by a Warbug impedance element, and the Bode plot of **18** reveals two time constants in contrast to all other examples.

Next, we tested **3**, **18**, **OPA** and **TC** in more corrosive media containing 3 wt% NaCl. The resulting parameters from potentiodynamic polarization measurements are shown in Table 2, and a selection of corresponding polarization curves, OCP vs. time, and Bode plots from EIS measurements are depicted in Figure 2. Compared to the evaluation in hard water, higher concentrations of corrosion inhibitors had to be employed to achieve a reasonable protection. However, for **TC**, **OPA** and **18** (entries 2, 3 and 5 in Table 2) efficiency is relatively low, even at high inhibitor concentrations of 64 and 100 mmol/L respectively (corresponding to 3–4.1 wt% of the acidic corrosion inhibitors). Carboxyphosphonic acid **3** clearly stands out, with excellent corrosion protection (entry 4 in Table 2). This is reflected by the time-dependent OCP in Figure 2A. For carboxyphosphonic acid **3,** the OCP decreased within a few hours to a strongly reduced value in comparison to the reference solution and all other tested inhibitors. As mentioned above, this decrease in OCP values can be interpreted as assembly of a high-quality protective layer on the metal surface. Both other phosphonic acids, **18** and **OPA**, do not obviously assemble to stable protective layers under these corrosive conditions. Again, precipitation of salts, most likely formed with corrosion products, prohibits regeneration of protective layers with these compounds, leading to a time-dependent development of OCP to a value only slightly anodic to the reference solution. **TC** shows a steady development of OCP to a significantly decreased value in comparison to the reference solution. Due to its good solubility, **TC** layers (although less protective than layers of **3**) get regenerated constantly. 

At lower chloride concentration (0.5 wt% NaCl), the amount of phosphonic acid inhibitors needed to achieve almost complete corrosion protection decreased significantly, to 40 mmol/L for **18** and 20 mmol/L for **3** (entries 6 and 7 in Table 2). **TC** also gained some efficiency in these less-corrosive conditions (entry 8 in Table 2). However, even at 64 mmol/L (corresponding to 3 wt%) inhibitor concentration, corrosion protection was still not complete. Full protection for **TC** was achieved only if the NaCl concentration was further decreased to 0.1 wt% aqueous NaCl (entry 9 in Table 2). At this point, it is unclear why the protective effect of **TC** was compromised so strongly by NaCl. In contrast, phosphonic acid **3** showed excellent corrosion protection in hard water and in aqueous NaCl solution. These findings are again supported by Bode plots of electrochemical impedance measurements, which confirm the high efficiency of carboxyphosphonic acid **3** as a layer-forming corrosion inhibitor in NaCl solutions (Figure 2C). The values of the Bode module in the low-frequency region are by far the largest for **3**. Bode plots reveal only one time constant, suggesting a charge-transfer-controlled corrosion process. The difference in size of the capacitive loops in the Nyquist plots is also extremely large among the compounds tested (see Appendix A, Appendix A).

#### 2.2.3. Gravimetric Evaluation of Anticorrosive Properties

As noted in the electrochemical evaluation above, the performance of layer-forming corrosion inhibitors can be time-dependent. Since many industrial applications require efficient corrosion inhibition over longer periods than those measured with our electrochemical studies, we included a long-term gravimetric assay with rectangular steel slides (S235JR) of 30 × 10 × 3 mm size. Due to the choice of test slides (S235JR) with intrinsically high corrosion resistance, and the limited sensitivity of gravimetric measurements, corrosive media were needed to increase the weight loss upon corrosion of the test slides to reliably detectable amounts within a reasonable timespan. We found 2% aqueous NaCl to be sufficiently corrosive to provide a weight loss of 186 mg within 12 weeks of incubation at room temperature and constant pH~8 (TEA/AcOH buffer) when no additional corrosion inhibitor was added. This amounts to a corrosion rate of 121 µm/y (Figure 3). All corrosion inhibitors tested were compared to this value, and the corresponding reduction in corrosion is given in Figure 3 (red bars) next to the corrosion rates (blue bars). We have used the TEA/AcOH buffer as a reference system, because the pH is in the same range as in our test solutions, and AcOH is known to have almost no anticorrosive properties at neutral or slightly alkaline pH values on iron and steel [52]. We focused on a selected set of readily-available phosphonic acids (**3**, **12**, **14** and **18**) for comparison. Compounds **12** (as a cyclic alkylphosphonic acid) and **14** (as a diphosphonic acid) were again included at this point to check the long-term performance of two derivatives, which do not belong to the class of carboxyphosphonic acids. As depicted in Figure 3, neither diphosphonic acid **14** or the reference compound **TC** showed good anticorrosive properties, with only 16% and 26% reduction, respectively. In contrast, the alkylphosphonic acids **OPA** and **12**, as well as carboxyphosphonic acid **18** showed reasonable anticorrosive effects with reductions around 60% (Figure 3). A remarkably strong protective effect was again observed for the carboxyalkylphosphonic acid **3**, which showed a reduction of 88%, confirming the results obtained by electrochemical methods. Additionally, further chip filter assays in 0.5% aqueous NaCl showed that **TC** fails to provide a reasonable anticorrosive effect (score 4 at 64 mmol/L inhibitor), whereas **3** gave excellent protection under the same conditions (score 0 at 20 mmol/L inhibitor). All three assays employed suggest that of the tested acidic corrosion inhibitors, only carboxyphosphonic acid **3** shows excellent anticorrosive properties in hard water and in the presence of chloride. We assume that **3** forms particularly dense layers on the metal surface and protects the surface from chloride attack, presumably by charge repulsion. Reven and Spiess have investigated the immobilization of carboxyphosphonic acids on ZrO_2_ and found that these compounds tend to form hydrogen-bonded multilayers on the surface [53]. Similar multilayers might be formed on steel surfaces, which would be an explanation for the outstanding performance observed with **3**. The strong anticorrosive effect of carboxyalkylphosphonic acid **3** in chloride media parallels observations of Benzakour et al., who observed similar effects in the comparison of a piperazine-based diphosphonic acid with the corresponding monophosphonic acid [54], and was also noted for alkyldicarboxylic acids [55]. An additional positive property of carboxyphosphonic acid **3** is good solubility even in hard water, which is limited for many other alkylphosphonic acids.

## 3. Discussion

In this work, we studied natural product-derived amphiphilic phosphonic acids as corrosion inhibitors for iron and steel in aqueous media at slightly alkaline pH. With respect to their preparation, various terpenoids and fatty acids containing double bonds or allylic alcohol motifs are attractive starting materials for the synthesis of phosphonic acids, either by the Pd-catalyzed Montchamp protocol or radical phosphorylation. Both reactions are scalable, and provide the desired phosphonic acids in one or two synthetic steps. Since starting materials such as fatty acids or terpenoids can be extracted from renewable plants such as feedstock, the resulting “semisynthetic” phosphonic acids might be commercially attractive and environmentally friendly substitutes for current corrosion inhibitors, which are typically products from an oil-based industrial value chain.

The results obtained show that amphiphilic phosphonic acids such as **3**, **7**, **12** and **18** have a comparable molar efficiency with respect to their anticorrosive properties as commercial acidic inhibitors, such as the tricarboxylic acid **TC**, in demineralized water. As their molecular weight is significantly lower than that of **TC**, phosphonic acids **3**, **7**, **12** and **18** have a significantly higher efficiency if atom economy is considered. However, the anticorrosive properties of most phosphonic acids tested were found to be strongly dependent on media effects and on their molecular structures. We observed the highest efficacy with long-chain carboxyphosphonic acids, such as **3** and **18**. While time-dependent anticorrosive properties were expected (phosphonic acids are known to assemble slowly into protective layers), the strong media effects which were observed were unexpected. We attribute the loss of anticorrosive effects with alkylphosphonic acids such as **12** or **OPA** in hard water and highly corrosive media to the limited solubility of salts with di- and trivalent cations from the media or corrosion products. This has important implications for real-world applications in metal working fluids, and restricts the use of such compounds to salt-free media, where several of our tested phosphonic acids have anticorrosive properties at least comparable to industrial benchmark inhibitors such as **TC**. Carboxyphosphonic acids such as **3** were found to be powerful corrosion inhibitors in corrosive media with high chloride content, as confirmed by a visual chip filter test on grey cast iron, a gravimetric test and electrochemical analysis on S235JR steel. From a mechanistic point of view, the protective effect of carboxyphosphonic acid **3** might be explained by efficient layer formation on the metal surface, and subsequent charge repulsion of the carboxy layer with chloride anions. It is remarkable that **TC** does not provide the same protective effect at high chloride concentration although it is a tricarboxylic acid. Carboxyphosphonic acid **3** is thus a particularly valuable corrosion inhibitor for iron and steel in contact with hard water or chloride-containing media. 

## 4. Materials and Methods

Full experimental details are given exemplarily for the synthesis of carboxyphosphonic acid **3** by Pd-catalyzed and **18** by radical hydrophosphonylation. Experimental procedures and analytical data for all other compounds can be found in the Appendix A.

### 4.1. NMR Spectroscopy

The ^1^H-, ^13^C- and ^31^P-NMR experiments were carried out at room temperature with a FourierHD 300 MHz, Avance III HD 400 MHz or Avance III HD 600 MHz spectrometer from Bruker Biospin GmbH (Ettlingen, Germany), calibrated against deuterated solvents or TMS. Additionally, HSQC, H,H-COSY and HMBC spectra were measured for peak assignment.

### 4.2. Mass Spectrometry

High-resolution mass spectra were measured with a MicrOTOF-Q with ESI source from Bruker Daltonik GmbH (Bremen, Germany).

### 4.3. Chip Filter Test

Evaluation of the anti-corrosive properties of the substances for iron was carried out according to DIN 51360-02-A, using an aqueous TEA-neutralized solution of the acidic corrosion inhibitors in either hard water (20 dGH: CaCl_2_ 340 mg/L, MgSO_4_ 60.0 mg/L) or 0.5% aqueous NaCl (pH 8.3 ± 0.5) at the appropriate concentration (Appendix A, see Appendix A). 1.5 eq TEA were added for each acidic proton in acidic corrosion inhibitors. As an example, carboxyphosphonic acid **3** contains 3 acidic protons, and thus 4.5 eq TEA were added as alkaline additive. A total of 2 mL of the appropriate test solution was incubated with 2.0 g of sieved grey cast iron turnings for 2 h, before rinsing and visual scoring according to DIN 51360-02-A. For data see Appendix A: Appendix A.

### 4.4. Gravimetric Corrosion Test

Rectangular S235JR steel slides (30 × 10 × 3 mm, obtained from Franz Krüppel GmbH+Co.KG (Krefeld, Germany) ingredients according to DIN EN 10025-2: C = 0.17%, Mn = 1.40%, P = 0.035%, S = 0.035%, N = 0.012%, Cu = 0.55%) were treated according to ASTM G1 C3.5, and cleaned in an ultrasonic bath (HCl 6 m; Urotropine 7 g/L) for 10 min and then dried in a nitrogen stream. Afterwards, the steel slides were immersed in test solutions for 12 weeks at room temperature, with regular substitution of evaporated water by addition of demineralized water. Test solutions were obtained by dissolving the appropriate acidic corrosion inhibitor in 2% aqueous NaCl and addition of 1.5 eq TEA for each acidic proton in acidic corrosion inhibitors. As an example, carboxyphosphonic acid **3** contains 3 acidic protons and thus 4.5 eq TEA were added as alkaline additive. The pH of the resulting test solutions was 8.3 ± 0.5. The corrosion rate *C_R_* in µm/y was calculated as follows:$$C_{R}=\frac{\Delta m}{A\;\cdot \;d\;\cdot \;t}365\;\cdot \;10^{9}$$
with *Δm* = weight loss in g, *A* = 840 mm^2^ (sample surface), *d* = 7940 kg/m^3^ (density of steel) and *t* = 84 d (immersion time). Each experiment was performed in triplicate.

### 4.5. Electrochemistry

Quadratic S235JR steel slides (30 × 30 × 2 mm, obtained from Franz Krüppel GmbH+Co.KG ingredients according to DIN EN 10025-2: C = 0.17%, Mn = 1.40%, P = 0.035%, S = 0.035%, N = 0.012%, Cu = 0.55%) for the electrochemical assays were treated according to ASTM G1 C3.5, and cleaned in an ultrasonic bath (HCl 6 m; Urotropine 7 g/L) for 10 min and then dried in a nitrogen stream. Potentiodynamic polarization curves were measured with a Gamry (Warminster, PA, USA) ParaCell setup. The samples were used as a working electrode, and were exposed to the electrolyte with an O-ring sealed opening with a diameter of 1 cm^2^. The counter electrode was graphite, and the reference electrode was Ag/AgCl in a 3 M KCl solution (SCE), which has a shift of +210 mV compared with a standard hydrogen electrode (SHE). Open-circuit measurements were taken over a time of 200 s. Impedance measurements were carried out using AC signals of a 10 mV potential perturbation, and the scanning frequency range was 100 kHz to 0.01 Hz with 5 point/decade. The polarization scan was between –100 mV and +100 mV against the open circuit potential, with a scan rate of 5 mV/s. The electrolyte was either hard water (20 dGH: CaCl_2_ 340 mg/L, MgSO_4_ 60.0 mg/L, conductivity at 25 °C: 861 μS/cm) or NaCl solution containing the corrosion inhibitor and TEA (final pH 8.3 ± 0.5). The polarization measurement was performed after 24 h exposure at room temperature. Data were processed with the Gamry (Warminster, PA, USA) Echem Analyst program (version 7.8.2). The percentage inhibition efficiency %*IE_corr_* was calculated as follows:$$IE_{corr}=\frac{I_{corr,\;\;\;uninhibited}-I_{corr,\;\;\;inhibited}}{I_{corr,\;\;\;uninhibited}}100\%$$
with *I_corr_* values from the uninhibited (hard water 20 dGH, pH 7.9–8.8) and inhibited solutions, respectively. Corrosion rates were calculated according to the literature [2]. 

### 4.6. Palladium-Catalyzed Hydrophosphinylation (General Procedure)

Pd_2_dba_3_ (75%, 0.01 eq, 0.1 mmol, 122 mg) and Xantphos (0.022 eq, 0.22 mmol, 127 mg) were dissolved in 10 mL of DMF. Hypophosphorous acid (50% in H_2_O, 2 eq, 20 mmol, 2.35 mL) and olefin, or allylic alcohol (1 eq, 10 mmol) were added. The reaction mixture was heated as indicated in the detailed procedures, filtered, and the solvent evaporated under reduced pressure. The residue was dissolved in 50 mL of 2 M aqueous HCl and 50 mL of EtOAc. The layers were separated, and the aqueous layer was extracted three times with 50 mL of EtOAc. The combined organic layers were washed with 50 mL of saturated NaCl solution and dried over Na_2_SO_4_. Removal of the solvent gave the crude product, which was oxidized or purified via RP column chromatography as specified.

#### Synthesis of 10-Carboxyundecylphosphonic Acid **3** [56]

10-Undecylenic acid **1** (82.8 g, 450 mmol, 1.0 eq) was treated according to the general procedure. After 16 h, quantitative conversion to the intermediate phosphinic acid was detected via mass spectrometry. The solution was cooled to 0 °C, and 30% aqueous H_2_O_2_ (102 g, 900 mmol, 2 eq) was added over 120 min. The reaction mixture was then heated to 110 °C for 60 min. The resulting mixture was filtered, and the solvent was evaporated under reduced pressure. The residue was treated with 250 mL 2 M aqueous HCl and 250 mL of EtOAc. The organic layer was separated and the aqueous layer extracted three times with 250 mL EtOAc. The combined organic layers were washed with 250 mL saturated NaCl solution and dried over Na_2_SO_4_. After crystallization from acetone and subsequent freeze drying, 119.5 g (449 mmol, 99%) of the title compound **3** were obtained as a colorless solid. ^1^H NMR (600 MHz, MeOD-*d*_4_) δ = 2.29 (dt, ^2^*J* = 20.4 Hz, ^3^*J* = 7.4 Hz, 2H, 10-H), 1.71–1.64 (m, 2H, 1-H), 1.60 (dqd, ^2^*J* = 12.1 Hz, ^3^*J* = 7.4, 6.6, 3.8 Hz, 4H, 3-H, 9-H), 1.41 (p, ^3^*J* = 6.6 Hz, 2H, 2-H), 1.36–1.28 (m, 10H, 4-H, 5-H, 6-H, 7-H, 8-H). ^13^C NMR (151 MHz, MeOD-*d*_4_) δ = 177.65 (C11), 34.86 (d, ^2^*J*_P,C_ = 23.2 Hz, C10), 31.75 (d, ^2^*J*_P,C_ = 16.6 Hz, C2), 30.51 (d, ^2^*J*_P,C_ = 3.2 Hz, (CH_2_)), 30.44 (CH_2_), 30.35 (d, ^2^*J*_P,C_ = 6.0 Hz, (CH_2_)), 30.23 (d, ^2^*J*_P,C_ = 8.7 Hz, (CH_2_)), 30.15 (CH_2_), 28.09 (d, ^2^*J*_P,C_ = 137.9 Hz, C1), 26.03 (d, ^2^*J*_P,C_ = 10.0 Hz, C9), 23.91 (d, ^2^*J*_P,C_ = 5.0 Hz, C3). ^31^P NMR (243 MHz, MeOD-*d*_4_) δ = 30.34. HRMS (ESI) *m*/*z*: [M − H]^−^ calculated for C_11_H_22_O_5_P: 265.1210, found 265.1210.

### 4.7. Synthesis of 10-Phosphonooctadecanoic Acid **18** [57]

To oleic acid **17** (90%, 1 eq, 159 mmol, 50.0 g) diethylphosphite (3 eq 478 mmol, 61.7 mL) was added under argon atmosphere. The solution was degassed for 30 min. Di-tert-butyl peroxide (0.1 eq, 16.0 mmol, 2.93 mL) was added. The reaction mixture was heated to 125 °C for 24 h. Another 0.1 eq Di-tert-butyl peroxide were added after 2 h, 16 h and 20 h. The reaction mixture was cooled to room temperature, and the volatile components were removed under oil pump vacuum. The residue was dissolved in 500 mL EtOAc, and washed twice with 500 mL saturated Na_2_CO_3_ solution. The organic layer was dried over Na_2_SO_4_ and filtered. Removal of the solvent under reduced pressure gave 65.2 g (155 mmol, 97%) of intermediate 10-(diethoxyphosphoryl)octadecanoic acid as a pale yellow oil. ^1^H NMR (300 MHz, CDCl_3_) δ = 4.15–4.02 (m, 4 H, 20-H), 2.27 (t, ^3^J = 7.6 Hz, 2 H, 9-H), 1.75–1.53 (m, 5 H, CH_2_), 1.53–1.18 (m, 31 H, CH_2_, 21-H), 0.87 (t, ^3^J = 6.7 Hz, 3 H, 18-H). ^13^C NMR (100 MHz, CDCl_3_) δ = 174.0 (d, ^10^J = 3.3 Hz, C10), 61.5 (C20), 60.3 (CH_2_), 36.1 (d, ^1^J = 137 Hz, C1), 34.5 (CH_2_), 34.1 (CH_2_), 32.0 (CH_2_), 29.9 (CH_2_), 29.7 (CH_2_), 29.6 (CH_2_), 29.5 (CH_2_), 29.4 (CH_2_), 29.3 (CH_2_), 29,1 (CH_2_), 27.8 (CH_2_), 27.7 (CH_2_), 25.1 (CH_2_), 24.9 (CH_2_), 22.8 (CH_2_), 16.6 (d, ^3^J = 5.8 Hz, C21), 14.3 (d, ^9^J = 22.5 Hz, C18). ^31^P NMR (162 MHz CDCl_3_) δ] = 35.11 (s), 35.06 (s). HRMS-ESI m/z: [M + H]^+^ calculated for C_22_H_45_O_5_P, 421.3077, found 421.3092.

This intermediate ester (1 eq, 143 mmol, 60.0 g) was dissolved in a mixture of 300 mL dioxane and 300 mL aqueous HCl (12 M, 3.2 mol). The reaction mixture was heated to reflux for 24 h. The organic solvent was removed under reduced pressure, and the residue was diluted with 600 mL water. The pH value was adjusted to 12 with solid NaOH. The aqueous layer was washed with each 600 mL EtOAc three times. The pH value was adjusted to 1 with concentrated aqueous HCl. The aqueous layer was extracted three times with 600 mL Et_2_O. The combined organic layers were then dried over Na_2_SO_4_ and filtered. Removal of the solvent under reduced pressure and coevaporating five times with 50 mL Et_2_O gave 47.6 g (130 mmol, 91%) of the title compound **18** as an amorphous, waxy solid. ^1^H NMR (300 MHz, CDCl_3_) δ = 7.61 (s_br_, 2 H, POOH), 2.34 (t, ^3^*J* = 7.1 Hz, 2 H, 9-H), 1.80–1.57 (m, 5 H, CH_2_), 1.57–1.21 (m, 24 H, CH_2_), 0.88 (t, ^3^*J* = 6.8 Hz, 3 H, 18-H). ^13^C NMR (100 MHz, CDCl_3_) δ = 180.1 (C10), 35.6 (d, ^1^*J* = 140 Hz, C1), 34.2 (CH_2_), 32.1 (CH_2_), 29.9 (CH_2_), 29.7 (CH_2_), 29.6 (CH_2_), 29.5 (CH_2_), 29.1 (CH_2_), 28.9 (CH_2_), 28.1 (CH_2_), 28.0 (CH_2_), 27.9 (CH_2_), 27.7 (CH_2_), 27.6 (CH_2_), 24.7 (CH_2_), 22.8 (CH_2_), 14.3 (C18). ^31^P NMR (162 MHz CDCl_3_) δ = 40.45 (s), 40.32 (s). HRMS-ESI *m/z*: [M − H]^−^ calculated for C_18_H_37_O_5_P, 363.2306, found 363.2296.

## 5. Conclusions

Natural products from renewable feedstocks, such as fatty acids or terpenoids, are attractive starting materials for “semisynthetic” phosphonic acids, which might be commercially attractive and environmentally friendly substitutes for current corrosion inhibitors. Carboxyphosphonic acid **3** was found to be a particularly good corrosion inhibitor, with a number of favorable properties such as anticorrosive effects in corrosive media with high chloride content and good hard water stability. Gravimetric tests and electrochemical analysis on S235JR steel revealed efficient layer formation on the metal surface and anodic corrosion protection.

## 6. Patents

As a result of the work reported herein, a patent application has been filed: WO/2021/165313, “Natural-product-derived phosphonic acids as acidic corrosion inhibitors”.

## Data Availability

Data are available from the authors upon request.

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
