# Peer review of "Natural Product-Derived Phosphonic Acids as Corrosion Inhibitors for Iron and Steel"

_molecules, 2022, doi:10.3390/molecules27061778_

Round 1
Reviewer 1 Report
P1 line 27. Define NACE.
P2 please remove the paragraph lines 69-77.
P3 line 111. The catalyst loading can be lowered a great deal: see Org. Chem. Front., 2019, 6, 2095 and US Pat. Appl 2011/0251314A1.
P3 line 97. Montchamp's J. Org. Chem. 2008, 73, 2292 should be added to reference 32.
P3 line 116. Allylic alcohols
P3 line 122. Steric hindrance
P3 line 128 and P4 line 137. Terpenoid. Please check for other occurrences.
P5 Scheme 4. Isomerization.
P6 Scheme 5. Did the authors consider hydrogenating compound 6 to compound 28? It would have been more efficient/economical.
P6 line 229. Is a specific molar quantity of corrosion inhibitor used? This seems to be potentially important but could not be found on P10 line 480 either. However, concentrations are mentioned on P7.
P7 line 284. Why abbreviate demineralized? Same on P9 Figure 3 in the caption.
P7 line 291. For a lack of regeneration
P11 line 417. The outstanding performance observed with 3.
P13 line 520. Do not capitalize dba to be consistent with scheme 1. Same comment in the Supporting Information.
P13 line 521. Hypophosphorous acid NOT hypophosphoric.
P13 line 529. 3 should be boldface
P14 line 548. 18 should be boldface
Supporting Information P4 general procedure. Allylic alcohol
Supporting Information reference 4 does not have a page number.
Toxicity is mentioned as an important parameter but there are no toxicity studies that are reported in the manuscript. Information about compound 3 and OPA would be welcome.
In Sci Finder, the LD50 for OPA is listed a 150-181 mg/kg. This is comparable to sodium nitrite which is used in small quantity as a food additive. Nonetheless, deriving corrosion inhibitors from renewable natural resources does not address any toxicity issue.
Overall, the manuscript is interesting, detailed, and generally well-written. Several assays were employed to evaluate the compounds that were synthesized. Compound 3 emerged as a potentially valuable corrosion inhibitor.
Publication is recommended after minor revisions outlined above.
Author Response
Reviewer #1:
P1 line 27. Define NACE.
Response: done
P2 please remove the paragraph lines 69-77.
Response: done
P3 line 111. The catalyst loading can be lowered a great deal: see Org. Chem. Front., 2019, 6, 2095 and US Pat. Appl 2011/0251314A1.
Response: sentence on page three was slightly modified and new reference 34 included
P3 line 97. Montchamp's J. Org. Chem. 2008, 73, 2292 should be added to reference 32.
Response: reference was included
P3 line 116. Allylic alcohols
Response: corrected in the complete manuscript
P3 line 122. Steric hindrance
Response: corrected
P3 line 128 and P4 line 137. Terpenoid. Please check for other occurrences.
Response: corrected in the complete manuscript
P5 Scheme 4. Isomerization.
Response: corrected
P6 Scheme 5. Did the authors consider hydrogenating compound 6 to compound 28? It would have been more efficient/economical.
Response: This is absolutely correct. However, since neither of the compounds 6, 27 and 28 turned out to be of particular interest for large scale application, we did not optimize their synthesis.
P6 line 229. Is a specific molar quantity of corrosion inhibitor used? This seems to be potentially important but could not be found on P10 line 480 either. However, concentrations are mentioned on P7.
Response: The specific concentrations of acidic corrosion inhibitors are listed in Table 2 in the supporting information. We have added a comment on pages 6 and 10.
P7 line 284. Why abbreviate demineralized? Same on P9 Figure 3 in the caption.
Response: corrected in the complete manuscript
P7 line 291. For a lack of regeneration
Response: corrected
P11 line 417. The outstanding performance observed with 3.
Response: corrected
P13 line 520. Do not capitalize dba to be consistent with scheme 1. Same comment in the Supporting Information.
Response: corrected
P13 line 521. Hypophosphorous acid NOT hypophosphoric.
Response: corrected
P13 line 529. 3 should be boldface
Response: corrected
P14 line 548. 18 should be boldface
Response: corrected
Supporting Information P4 general procedure. Allylic alcohol
Response: corrected
Supporting Information reference 4 does not have a page number.
Response: This is a bit uncommon, but the article has no page numbers assigned. The journal provides article numbers instead, which we have now included in the reference.
Toxicity is mentioned as an important parameter but there are no toxicity studies that are reported in the manuscript. Information about compound 3 and OPA would be welcome. In Sci Finder, the LD50 for OPA is listed a 150-181 mg/kg. This is comparable to sodium nitrite which is used in small quantity as a food additive. Nonetheless, deriving corrosion inhibitors from renewable natural resources does not address any toxicity issue.
Response: We have erased the term “non-toxic” from the introduction. As the reviewer points out, similar alkylphosphonic acids like OPA have low toxicity. However, we cannot provide toxicity data for 3 and 18 at this point.
Overall, the manuscript is interesting, detailed, and generally well-written. Several assays were employed to evaluate the compounds that were synthesized. Compound 3 emerged as a potentially valuable corrosion inhibitor.
Publication is recommended after minor revisions outlined above.
Reviewer 2 Report
In this study, organic corrosion inhibitors based on phosphonic acids from renewable resources were evaluated by chip-filter assay, potentiodynamic polarization measurements, electrochemical impedance measurements and gravimetry for corrosion protection of iron and steel in aqueous environment at slightly alkaline pH. The study achieves certain originality. However, I think that the format of this study is confused, which should be improved in these aspects.
- The writing sequence of this study is in the sequence: 1. Introduction, 2. Results, 3. Discussion and 4. Materials and Methods. However, the routine order of research paper is 1. Introduction, 2. Materials and Methods, 3. Results, 4. Discussion and 5. Conclusions.
- Some sequences should be deleted. For example, in my opinion, the paragraph of “The introduction should briefly place the study in a broad context and highlight why it is important. ………..See the end of the document for further details on references” (in Introduction) should be deleted.
- The used corrosive medium in this study is hard water (20dGH) and demin. water. I wonder their components and conductivity. In my opinion, sodium chloride should be selected instead of hard water (20dGH) and demin. water in order to obtain accurate results.
- The Nyquist of EIS should be provided
Author Response
Reviewer #2:
In this study, organic corrosion inhibitors based on phosphonic acids from renewable resources were evaluated by chip-filter assay, potentiodynamic polarization measurements, electrochemical impedance measurements and gravimetry for corrosion protection of iron and steel in aqueous environment at slightly alkaline pH. The study achieves certain originality. However, I think that the format of this study is confused, which should be improved in these aspects.
The writing sequence of this study is in the sequence: 1. Introduction, 2. Results, 3. Discussion and 4. Materials and Methods. However, the routine order of research paper is 1. Introduction, 2. Materials and Methods, 3. Results, 4. Discussion and 5. Conclusions.
Response: We followed the instructions of the journal, which are given in the journal template for “Molecules”. However, we have also no objections against a change in the order of chapters. The editorial office might want to decide on this issue.
Some sequences should be deleted. For example, in my opinion, the paragraph of “The introduction should briefly place the study in a broad context and highlight why it is important. ………..See the end of the document for further details on references” (in Introduction) should be deleted.
Response: done
The used corrosive medium in this study is hard water (20dGH) and demin. water. I wonder their components and conductivity. In my opinion, sodium chloride should be selected instead of hard water (20dGH) and demin. water in order to obtain accurate results.
Response: This is most likely a misunderstanding: We have used hard water (20dGH) and sodium chloride solutions of different concentrations (0.1-3 wt%) as media for chip-filter assays (see Table 2, SI) and electrochemical evaluation (see Tables 1-2 and Figures 1-2 in the manuscript). Only one experiment with OPA (Table 1, entry 9 and brown curves in Fig. 1) has been performed in demineralized water. Through the relatively high concentrations of the OPA/TEA salt (19 mmol/L corresponding to 0.7 wt% acidic corrosion inhibitor) it serves as an electrolyte in this case. The components of hard water 20dGH are listed in the experimental section in section 4.3. and 4.5. We have also added the conductivity for hard water (20dGH) to section 4.5. For gravimetric tests, 2% aqueous sodium chloride solution was used as solvent (see Materials and Methods, section 4.4.).
The Nyquist of EIS should be provided
Response: We have included the Nyquist plots in the supporting information.
Reviewer 3 Report
In my opinion, the paper is well structured and organized.
It could be publishe by the Journal.
The only question is Why impedance data are not described?
Spectra contain interesting results disclosing differences in the mechanism of protection and their description could be beneficial for the manuscript
Author Response
Reviewer #3:
In my opinion, the paper is well structured and organized.
It could be published by the Journal.
The only question is Why impedance data are not described?
Spectra contain interesting results disclosing differences in the mechanism of protection and their description could be beneficial for the manuscript
Response: We have added a discussion of the impedance data in lines 338-345 and 384-390.
Round 2
Reviewer 2 Report
After revision, the manuscript has been improved significantly. However, I think that the paper should be improved in these aspects:
- The place between the reference number and a comma or a full stop at the end of the sentence should be considered. In general, the reference number is before a comma or a full stop. Therefore, the sentence of “…..to NACE (National Association of Corrosion Engineers).[1]” should be changed into “…..to NACE (National Association of Corrosion Engineers) [1].”
- The chemical compositions or the producing company of S235JR steel should be provided.
- The relation between the capacitive loops or the moduli (|Z|) value with the corrosion resistance should be given. For example, the larger radius of capacitive loops in the Nyquist plots indicates the better corrosion resistance. The larger the moduli (|Z|) value is, the better the corrosion resistance is. Some papers should be cited, for example, Xiaoting Shi, Yu Wang, Hongyu Li, et al., Corrosion resistance and biocompatibility of calcium-containing coatings developed in near-neutral solutions containing phytic acid and phosphoric acid on AZ31B alloy, 2020, 823, 153721; M. Soleymanibrojeni, H. W. Shi, I. I. Udoh, et al., Microcontainers with 3-amino-1,2,4-triazole-5-thiol for Enhancing Anticorrosion Waterborne Coatings for AA2024-T3, Progress in Organic Coatings, 137 (2019) 105336).
Author Response
Reviewer #2, 2nd revision:
The place between the reference number and a comma or a full stop at the end of the sentence should be considered. In general, the reference number is before a comma or a full stop. Therefore, the sentence of “…..to NACE (National Association of Corrosion Engineers).[1]” should be changed into “…..to NACE (National Association of Corrosion Engineers) [1].”
Response: done
Some sequences should be deleted. For example, in my opinion, the paragraph of “The introduction should briefly place the study in a broad context and highlight why it is important. ………..See the end of the document for further details on references” (in Introduction) should be deleted.
Response: done
The chemical compositions or the producing company of S235JR steel should be provided.
Response: The information has been added to sections 4.4 and 4.5.
The relation between the capacitive loops or the moduli (|Z|) value with the corrosion resistance should be given. For example, the larger radius of capacitive loops in the Nyquist plots indicates the better corrosion resistance. The larger the moduli (|Z|) value is, the better the corrosion resistance is. Some papers should be cited, for example, Xiaoting Shi, Yu Wang, Hongyu Li, et al., Corrosion resistance and biocompatibility of calcium-containing coatings developed in near-neutral solutions containing phytic acid and phosphoric acid on AZ31B alloy, 2020, 823, 153721; M. Soleymanibrojeni, H. W. Shi, I. I. Udoh, et al., Microcontainers with 3-amino-1,2,4-triazole-5-thiol for Enhancing Anticorrosion Waterborne Coatings for AA2024-T3, Progress in Organic Coatings, 137 (2019) 105336).
Response: A passage explaining the correlation of Bode moduli and size of capacitive loops was included (lines 335-336) along with the two new references 50 and 51 as suggested.